# OpenReview forum: "AVI-Bench: Toward Human-like Audio-Visual Intelligence of Omni-MLLMs"
_ICLR.cc/2026/Conference — Submitted to ICLR 2026_

### Official Review · Reviewer_qguB · 2025-10-27

**Soundness:** 3
**Presentation:** 3
**Contribution:** 3
**Rating:** 6
**Confidence:** 4

**Summary:**

This paper presents AVI-Bench, a benchmark to evaluate Omni-MLLMs across omni-modal perception, understanding, and reasoning ability. To explore the Omni-MLLM's generalization towards unfamiliar sensory inputs, they further propose AVI-Bench-PriSe, an extension of AVI-Bench that focuses on unfamiliar-domain audio-visual inputs. AVI-Bench contains over 5000 samples and was evaluated on more than 10 open-source and closed-source models, which showed that perception and understanding might be bottlenecks for reasoning, and on unfamiliar input, there is still a large room for improvement.

**Strengths:**

1. **Comprehensive tasks.** AVI-Bench incorporate 13 task types across categories such as understanding, perception, and reasoning, allowing for a thorough evaluation of Omni-MLLM’s understanding across different aspects.

2. **Well-grounded tasks.** Most tasks in AVI-Bench require comprehension across more than one modality, enabling a comprehensive assessment of multimodal collaboration capability for Omni MLLMs. In particular, it can also evaluate the ability to understand multi-segment audio and multiple images.

3. **4-level Classification Scheme** This paper proposes a 4-level classification scheme, which is reasonable and insightful, provides good metrics for evaluating Omni-MLLM.

**Weaknesses:**

1. **Discussion with related work.** There is some related work on evaluating Omni-MLLM which are not compared and discussed with. The author should compare AVI-Bench with these works and show the main differences.
[1] Hong, Jack, et al. "Worldsense: Evaluating real-world omnimodal understanding for multimodal llms." arXiv preprint arXiv:2502.04326 (2025).
[2] Daily-Omni: Zhou, Ziwei, Rui Wang, and Zuxuan Wu. "Daily-Omni: Towards Audio-Visual Reasoning with Temporal Alignment across Modalities." arXiv preprint arXiv:2505.17862 (2025).
[3] Video-Holmes: Cheng, Junhao, et al. "Video-Holmes: Can MLLM Think Like Holmes for Complex Video Reasoning?." arXiv preprint arXiv:2505.21374 (2025).

**Questions:**

1. **Adding more models.** Suggest adding more open-source Omni-MLLMs in this benchmark in Table 3 for a more comprehensive evaluation.
[1] Mini-Omni2: Xie, Zhifei, and Changqiao Wu. "Mini-omni2: Towards open-source gpt-4o with vision, speech and duplex capabilities." arXiv preprint arXiv:2410.11190 (2024).
[2] Lyra: Zhong, Zhisheng, et al. "Lyra: An efficient and speech-centric framework for omni-cognition." Proceedings of the IEEE/CVF International Conference on Computer Vision. 2025.
[3] MiniCPM-o 2.6: Yuan Yao, et al. "Minicpm-o 2.6: A gpt4o level mllm for vision, speech, and multimodal live streaming on your phone."

---

> ### Author Response · Authors · 2025-11-25
> **Response to Reviewer qguB**
>
> We thank your thoughtful response and we hope to address your concerns point by point in the following replies, and we appreciate your patience, time, and constructive comments.
>
> ---
> ### 1. Discussion with related works
>
> We thank your valuable suggestions and for providing these important and inspiring related works. We compare our work with these studies as follows.
>
> - Worldsense [1] focuses on the understanding capabilities of Omni-MLLMs in real-world scenarios, emphasizing the model's ability to simultaneously comprehend both audio and visual content in videos.
>
> - Daily-Omni [2] explores the reasoning performance of Omni-MLLMs in relatively longer videos (30–60 seconds), highlighting the importance of audio-visual temporal alignment.
>
> - Video-Holmes [3] aims to assess whether MLLMs can perform complex video reasoning similar to human experts. Although this dataset investigates the model's ability to process audio inputs for supplementary understanding, its primary focus remains on visual content as the core reasoning task, as the data is derived from suspense short films.
>
> In contrast, AVI-Bench is designed to explore the **human-like audio-visual intelligence** of Omni-MLLMs, relative to general human abilities. It is therefore structured as a **cognitively-grounded benchmark** encompassing sensation, perception, understanding, and reasoning. We also introduce a novel audio-visual **intelligence categorization taxonomy** with corresponding metrics, allowing us to *quantify the intelligence levels* of Omni-MLLMs across four dimensions: Task-, Modality-, Stage-, and Domain-Adaptive.
>
> Moreover, while these benchmarks primarily focus on QA with multiple-choice options, AVI-Bench offers a **broader range of data formats** from diiferent tasks, including QA with multiple-choice options, *localization and tracking with bounding boxes*, and open-ended questions, as detailed in Table 1. We hope that these diverse task formats will provide a more comprehensive evaluation of Omni-MLLM's human-like audio-visual intelligence.
>
> We again greatly appreciate your suggestions and supplements. We promise to include the discussion with related works in our revised version.
>
> ---
> ### 2. Results on more models
>
> We thank you for the valuable suggestion. We have included the evaluations for Mini-Omni2, Lyra, and MiniCPM-o below.
>
> | | | Perception|  | | | | Understand |  | | |Reasoning  | | | | |Primitative Sensation | | | | avg. |
> |-|-|-|-|-|-|-|-|-|-|-|-|-|-|-|-|-|-|-|-|-|
> | **Omni-MLLMs** | **Params** | **AMIC** | **VMIC** | **AVL**  | **AVM**  | **avg.** | **VAR**  | **AVR**  | **AVC**  | **avg.** | **AVH**  | **VAH**  | **AVQA** | **AVLG** | **avg.** | **ASQA** | **VSQA** | **AVSQA** | **avg.** |
> | | | | | | | | | | | | | | | | | | | | |
> |Mini-Omni2 | 0.5B | 00.00 | 07.32 | 03.49 | 19.10 | 07.48 | 01.26 | 03.09 | 21.58 | 08.64 | 32.49 | 34.77 | 33.14 | 00.00 | 25.14 | 00.50 | 09.35 | 06.44 | 05.43 |11.67  |
> |Lyra-Base | 9B | 10.14 | 38.70 | 04.36 | 48.40 | 25.40 | 08.27 | 21.52 | 18.61 | 16.13 |  74.00 | 58.40 | 46.17 | 00.46 | 44.76 | 08.60 | 28.39 | 03.22 | 13.40 | 24.92 |
> |MiniCPM-o | 8B | 17.00 | 56.33 | 08.99 | 45.60 | 31.98 | 25.52 | 20.58| 19.88 | 21.99 | 82.80 | 64.40 | 61.41 | 05.54 | 53.54 | 08.77 | 29.95 | 07.38 | 15.37 | 30.72
>
> In addition, AVI-Bench includes a corresponding testing toolkit with a model reasoning entry point for convenient testing.
>
> > [1] Worldsense. arxiv.org/abs/2502.04326
> [2] Daily-Omni. arxiv.org/abs/2505.17862
> [3] Video-Holmes. arxiv.org/abs/2505.21374
> [4] Mini-Omni2. arxiv.org/abs/2410.11190
> [5] Lyra. arxiv.org/abs/2412.09501
> [6] MiniCPM-o. arxiv.org/abs/2405.17220

---

### Official Review · Reviewer_3GeF · 2025-10-29

**Soundness:** 2
**Presentation:** 2
**Contribution:** 1
**Rating:** 2
**Confidence:** 4

**Summary:**

This work introduces AVI-Bench, a cognitively inspired benchmark designed to assess Omni-MLLMs across perception, understanding, and reasoning capabilities. The benchmark contains over 5K question–answer pairs covering 14 different tasks, grouped into perception, understanding, reasoning, and primitive sensation. To further approximate each Omni-MLLM’s performance in terms of human-like intelligence, the authors propose four levels of metrics to characterize audio-visual intelligence: task-adaptive intelligence, modality-adaptive intelligence, stage-adaptive intelligence, and domain-adaptive intelligence.

**Strengths:**

- Designing the benchmark in a cognitively inspired manner is an interesting direction for understanding the behavior of Omni-MLLMs from a human-like perspective.

**Weaknesses:**

- The contribution in constructing the dataset appears limited. As mentioned in Section A.6.1, most of the tasks are directly taken from existing benchmarks and only reformatted. The tasks newly created by the authors also overlap at a high level with existing ones. In addition, ASQA was previously proposed in AV-Odyssey Bench: Can Your Multimodal LLMs Really Understand Audio-Visual Information? (Arxiv, 2024).
While combining existing tasks from different benchmarks can be meaningful, the current scale of AVI-Bench, as shown in Table 1, is relatively small—insufficient to fully support claims of modeling human cognitive processes. What is the advantage of using subsets of existing benchmarks within AVI-Bench, compared to using the full datasets from each benchmark to evaluate Omni-MLLMs?

- In L330, the authors mention that PandaGPT-7B and PandaGPT-13B perform poorly in perception but well in understanding, which limits reasoning. However, perception should inherently affect understanding, as accurate understanding depends on accurate perception. Therefore, the categorization into four different tasks (perception, understanding, reasoning) may be somewhat entangled.

- Regarding Equation 1, each task uses a different evaluation metric with distinct meanings and scales. Is it appropriate to simply average all quantitative results across tasks? Were any normalization, weighting, or other methods applied to ensure consistency across tasks?

- Equation 2 also raises concerns. A and V represent performance on audio-only and visual-only dominant tasks, respectively, but why does A+V equal zero? Are A and V averages over tasks for each modality? Moreover, since the number of tasks and metric scales vary, is it valid to average them all? Additionally, from a cognitive perspective, the order should be perception → understanding → reasoning. Perception and understanding should not be treated as equivalent in priority.

**Questions:**

### Questions and Suggested Experiments

- Providing recommendations or experiments that could improve model performance in terms of human-like intelligence would improve this work.

### Minor Questions and Suggestions

- In Figure 1, for the visual-reference audio retrieval task, why is the correct answer “no” instead of selecting one option?

---

> ### Author Response · Authors · 2025-11-25
> **Response to Reviewer 3GeF (Part 1/3)**
>
> We sincerely thank you for acknowledging the design of our cognitive-grounded benchmark and for offering valuable feedback. Your insights will certainly help us improve the organization of our content and more clearly emphasize our contributions.
>
> We now address each of your concerns point by point in the following. Hope you will reconsider your evaluation if our responses are effective to you. We greatly appreciate your time, patience, and thoughtful feedback.
>
> ---
> We include citations here for better indexing and appreciate your understanding.
> > [1] General-Bench. ICML'25 Oral\
> [2] SAVE-Bench. ICML'24\
> [3] MMT-Bench. ICML'24\
> [4] AVTrustBench. ICCV'25\
> [5] AVHBench. ICLR'25\
> [6] OmnixR. ICLR'25\
> [7] AV-Odyssey. arXiv'24\
> [8] Q-Bench. ICLR'24\
> [9] WorldQA. arXiv'24\
> [10] OmniBench. NeurIPS'25\
> [11] AVCaps. IEEE OJSP'25\
> [12] WorldSense. arXiv'25\
> [13] Humanity's Last Exam. arXiv'25\
> [14] PandaGPT, Sec 4: Capabilities. TLLM'23
>
> ---
> ### 0. Contribution clarification
> Before addressing your concerns, we would like to clarify our contributions. Our goal is not to:
> - Construct millions or billions of data samples to model human cognitive processes.
> - Introduce multiple new tasks to evaluate the performance of Omni-MLLMs.
>
> Instead, we use repurposed (a widely accepted approach [1-6]) and new data to create a cognitive-grounded benchmark for evaluating the human-like audio-visual intelligence (AVI) of Omni-MLLMs:
> - **Framework Design**: AVI-Bench is the **first benchmark grounded in human cognition** to assess the AVI of Omni-MLLMs. Our four-stage evaluation framework (Sensation, Perception, Understanding, and Reasoning) provides a unified, interpretable structure for benchmarking.
> - **Four-Level Taxonomy and Metrics**: The taxonomy evaluates intelligence across four levels: Task-, Modality-, Stage-, and Domain-Adaptive. It is the **first taxonomy to categorize human-like AVI** in Omni-MLLMs.
> - **Insightful Findings**: Through evaluations of 28 models, we identify seven novel, critical observations (e.g., modality imbalance, stage-specific bottlenecks) that **reveal current limitations** and **suggest promising research directions**.
>
> We acknowledge your point about the importance of (1) large-scale datasets and (1) introduction of new tasks for advancing Omni-MLLMs. However, we believe our contributions above provide valuable insights that can inform future works focused on (1) and (2).
>
> ---
> ### 1. Limited contributions, most tasks are from existing benchmarks and only reformatted
> - 62% of the data consists of newly annotated samples, and 100% of the data underwent manual checks and reformatting for a more unified and efficient evaluation process.
> - While developing new tasks is crucial for advancing the audio-visual community, this work focuses on benchmarking the AVI of Omni-MLLMs using well-established, canonical, and thoroughly studied tasks from the audio-visual and Omni-MLLM communities.
> - We will revise the statement of our contributions in `Sec 1. Introduction` for better clarity.
>
> ---
> ### 2. ASQA was previously proposed in AV-Odyssey
> - The primary goal of AVI-Bench-PriSe is to assess the adaptive capabilities of Omni-MLLMs beyond commonly used tasks in large model training, such as VQA and AQA.
> - We **introduce AVSQA**, a novel and essential task that evaluates Omni-MLLMs' ability to process low-level audio-visual information. As highlighted in our benchmark's cognitive-grounded design, which you has kindly acknowledged (we thank again), we specifically include audio-dominated (ASQA) and visual-dominated (VSQA) tasks to **calculate the modality-adaptive score**.
> - We appreciate your reminder and will add the relevant reference[7-8] in `line 258~260` of `Sec 3.2 Stages and Tasks`.
>
> ---
> ### 3. What is the advantage of using subsets?
> - AVI-Bench is not just a dataset but primarily a benchmark that introduces a consistent evaluation protocol and a novel taxonomy of human-like AVI, offering clear guidance for future research.
> - While evaluating on full benchmarks remains valuable for assessing performance in specific aspects, the scale of existing benchmarks make exhaustive evaluation impractical due to significant demands on **time, computational resources, and environmental sustainability**.
> - **Using repurposed subsets is a common and effective strategy in modern benchmarking to enable efficient evaluation**. Our approach follows this practice, aligning with recent works such as **General-Bench (ICML'25 Oral)** [1], **SAVE-Bench (ICML'24)** [2], **MMT-Bench (ICML'24)** [3], **AVTrustBench (ICCV'25)** [4], **AVHBench (ICLR'25)** [5], and **OmnixR (ICLR'25) [6]**.
> - As noted in `Sec A.3.2 Statistical Significance`, we further validate that our subset-based evaluation yields statistically significant and reliable results.
> - We thank the reviwer to provide this concern, we will add the above in `Sec A.6 Pipeline and Quality Control` for better clarification.

---

> ### Author Response · Authors · 2025-11-25
> **Response to Reviewer 3GeF (Part 2/3)**
>
> ### 4. Small size and difficulty modeling human cognitive processes
> - (1) AVI-Bench is **designed to evaluate human-like AVI, instead of training models to build human cognitive processes** with large dataset. For example, WorldQA [9] with 1,007 QA pairs aims to push the boundaries of multimodal world models, and WordSense [12] with 3,172 QA pairs evaluates real-world scenarios, acknowledging that world knowledge is too complex to model with thousands of samples. Notably, the HLE [13] benchmark uses just 2,500 samples as "Humanity's Last Exam".
> - (2) AVI-Bench contains **5,864 samples, which is larger than many other benchmarks**: WorldQA [9] (1,007), OmniBench [10] (1,142), OmniXR [6] (1,800), AVCaps [11] (2,061), WordSense[12] (3,172), AV-Odyssey [7] (4,555), and AVHBench [5] (5,302).
> - (3) The effectiveness of **AVI-Bench to reflect human-like AVI is evident from Table 9**, which shows the performance comparision of human and Gemini-2.5-pro. Notably, humans achieve consistently high performance on AVI-Bench (avg. 92.6), far surpassing Gemini-2.5-Pro (49.6) and others. Compared to MMMU-Bench (human: 88.6, MLLM: 69.1), AVI-Bench shows higher human scores (92.6 vs. 88.6) but lower MLLM performance (49.6 vs. 69.1), indicating that **AVI-Bench is more aligned with normal human capability, and poses a greater challenge for MLLMs**.
> - (4) We validate the **statistical significance of the evaluation results** on our dataset in `Sec A.3.2 Statistical Significance`.
> - We appreciate your insightful concerns. We will add the analysis in points (1 & 2) to a new section, `Dataset Scale Discussion,` in the supplementary material, and include points (3 & 4) in `Sec 5.5 Comparison of Intelligence`.
>
> ---
> ### 5. Inconsistent behavior in perception and understanding for PandGPT
> - A model's behavior is largely determined by its training data. For example, PandaGPT [14] exhibits poor performance on the audio-visual localization task, suggesting it may not be optimized for strong grounding capabilities.
> - Instead, as indicated in their *paper [14], the model appears to be primarily trained on understanding-level tasks*, resulting in **behavior inconsistent with human cognition** across perception and understanding stages.
> - This further underscores the importance of hierarchical evaluation across different cognitive stages, and **reveals future directions for advancing human-like AVI**.
> - We thank your kind reminder and we will incorporate the above analysis into Observation 2 in `Sec 4.2 Results Analysis and Observations`.
>
> ---
> ### 6. Cognitive-grounded dataset design is entangled
> - As you noted, "perception may inherently affect understanding." This is precisely why a cognitive-grounded design is important: **traditional evaluations struggle to quantify fine-grained performance** when perception, understanding, and reasoning are entangled in a single benchmark.
> - To address this, we introduce distinct stages that disentangle these capabilities. In fact, as summarized in the section 5 and figure 6 of MMMU-Bench, 35% of reasoning errors stem from perception failures. This underscores the value of evaluating each cognitive stage separately, **enabling fine-grained, quantifiable assessment** of a model's abilities, with each stage focusing on its specific cognitive function.
> - We recognize your concern and we will add the above analysis into `Sec A.2.5 Human Performance`.
>
> ---
> ### 7. How task consistency is maintained in Equation 1
> - AVI-Bench employs **carefully designed and normalized metrics** to provide a unified evaluation framework on a consistent scale.
> - For instance, for tasks like AMIC and VMIC that require counting the number of instances, we compute both a semantic matching score and a counting error. Since the counting error is measured by RMSE (cf. Eqs. 13–16), we apply a normalized function `1-tanh(*)` to map the counting error (lower is better within `[0,+∞)`) to counting score (higher is better within `(0, 1])`) (cf. Eq. 17). This ensures alignment with the scale of other metrics.
> - Other metric designs are detailed in Section A.5 Metrics. For example, retrieval tasks incorporate penalization and confidence adjustment (cf. Eqs. 32–36) to mitigate the impact of answer repetition or verbatim copying, enabling more accurate evaluation.
> - We acknowledge your concern and will incorporate the above explanations into `Sec 3.4 Evaluation Metrics` with a cross-reference to `Sec A.5 Metrics` in the supplementary material.

---

> ### Author Response · Authors · 2025-11-25
> **Response to Reviewer 3GeF (Part 3/3)**
>
> ### 8. What is the meaning of A and V? Why A+V = 0?
> > "... A and V represent performance on audio-only and visual-only dominant tasks ... "
> - More precisely, A and V denote the average performance on **audio-dominant tasks** and **visual-dominant tasks**, respectively (line 401). Audio-dominant tasks include AMIC, VAR, AVH, and ASQA; visual-dominant tasks include VMIC, AVR, VAH, and VSQA (line 166). These tasks are **NOT** audio-only or visual-only.
>
> > "Are A and V averages over tasks for each modality?"
> - Yes. Similar to Eq. 1, we compute A as the average score over the four audio-dominant tasks and V as the average over the four visual-dominant tasks. We will clarify this in the revised manuscript.
>
> > "Why A + V = 0 ?"
> - In Eq. 2 (line 406), we use the term $\frac{|A-V|}{|A+V|}$. **To avoid division by zero**, we define $\Delta_m = 2$ when $A + V = 0$.
> - In the most extreme case, a model may perform very poorly on audio-visual tasks while still possessing an exceptionally strong and trustworthy capability, for example a highly capable language-only model. Its predictions on audio-visual tasks are effectively assigned a score of zero, as the model knows it cannot see or hear.
>
> > "The number of tasks and metric scales vary. Is it valid to average them all?"
> - As stated in line 166, both groups contain exactly four tasks, ensuring a **balanced comparison**.
> - Furthermore, as explained in Re#7 and `Sec A.5 Metrics`, all task scores are **normalized to the same scale** before averaging, making the aggregation valid.
>
> We thank you for raising these concerns and will incorporate the above discussion into `Sec 5.1~5.2`, with a cross-reference to `Sec A.5 Metrics` in the supplementary material.
>
> ---
> ### 9. The concern about the stage order or priority
> > "the order should be perception → understanding → reasoning. Perception and understanding should not be treated as equivalent in priority."
> - We do **NOT** claim that perception and understanding are equivalent in priority.
> - For the Stage-Adaptive Intelligence metric (cf. Sec 5.3), we use $\frac{|R_P - R_U|}{R_P + R_U}$ because we have already computed per-task scores.
> - Based on **quantitative results across 28 omni-MLLMs**, we observe that better understanding does not always lead to better perception. As discussed in Rebuttal #5, some models exhibit inconsistences due to unbalanced training data, resulting in varying strengths at the perception or understanding level. Certain models excel at perception tasks, while others perform better on understanding tasks.
> - However, **strong reasoning requires proficiency in both**. As stated in Observation 2: "both perception and understanding are essential for enhancing cross-modal reasoning." This is precisely why we propose a Stage-Adaptive approach: a model demonstrating human-like audio-visual intelligence should achieve high performance in both stages, rather than showing a bottleneck in one.
> - Our intelligence evaluation metrics are **grounded in real, empirical findings from experiments**, rather than being designed based on intuition and then used to interpret results.
> - We thank you for this critical feedback and will incorporate the above discussion into `Sec 5.3 Stage-Adaptive` to improve clarity.
>
> ---
> ### 10. Recommendations or experiments to improve model performance
> We agree with you that model performance could be improved in terms of human-like intelligence. Accordingly, we have proposed the following suggestions in our observations:
> - Enhance synergy across tasks at different levels.
> - Both perception and understanding are essential for reasoning.
> - Future work should aim for balanced audio-visual intelligence rather than specializing in a single modality.
> - Fine-grained perceptual capabilities need to be strengthened, which is common in humans but often missing in current models.
>
> Additionally, experiments on multi-stage baselines (cf. Sec A.2.3) and different modality encoders (cf. Sec A.2.4) revealed some insight:
> - upgrading the language backbone (i.e., the language intelligence) can improve audio-visual intelligence.
> - better modality encoders leads to better audio-visual intelligence.
>
> We thank your reminder and will incorporate the above into a new subsection, `Sec A.2.6 Discussions`, in `Sec A.2 Additional Experiments` to better draw the reader's attention.
>
> ---
> ### 11. why not selecting an option in Figure 1
> - We used "no" because, for this data sample, there is no audio that corresponds to the given image.
> - We appreciate you points out this and we will use "none" or an empty list "[]" in the revised version for a clearer visualization.

---

### Official Review · Reviewer_Drp2 · 2025-10-31

**Soundness:** 3
**Presentation:** 3
**Contribution:** 3
**Rating:** 6
**Confidence:** 3

**Summary:**

This paper presents AVI-Bench, a cognitively inspired benchmark for evaluating audio-visual intelligence in Omni-Multimodal LLMs such as GPT-4o and Gemini. It assesses models across three stages: Perception, Understanding, and Reasoning, as well as a Primitive Sensation stage for unfamiliar low-semantic inputs. The authors also evaluate various open-sourced and commercial models, revealing weaknesses in audio reasoning and cross-modal grounding.

**Strengths:**

- Comprehensive and cognitively grounded design: Mirrors human cognition and enables fine-grained diagnosis of model abilities.
- Extensive experimental coverage: Evaluates 28 models  (both open-sourced and commercial) across diverse tasks, giving strong empirical credibility.
- Insightful taxonomy and analysis: The four-level framework offers a clear lens to interpret model progress toward human-like intelligence.

Therefore, I believe it is a useful benchmark for audio-visual llms, even though similar benchmarks have been preoposed before.

**Weaknesses:**

- Considering similar benchmarks have been proposed, the main contribution is just an advanced benchmark for omni-MLLMs, not a pioneer in this field.
- Although the proposed AVI-Bench can evaluate the omni models under different modalities, it still cannot serve as a replacement of single modality evaluation, such as MMMU for image understanding.

**Questions:**

na

---

> ### Author Response · Authors · 2025-11-25
> **Response to Reviewer Drp2**
>
> We sincerely thank you for the constructive feedback on our work, and greatly appreciate the recognition of comprehensive and cognitively grounded benchmark design, extensive experiments, insightful taxonomy and analysis.
>
> We would like to address the concerns from the following points.
>
> ---
> ### 1. Novelty and Contributions
>
> We appreciate the your feedback. Unlike previous benchmarks that evaluate audio-visual capabilities using individual assessments for audio or visual tasks, or limited audio-visual joint tasks (which may lack fine-grained audio-visual localization and tracking), we would like to reiterate the novelty and contributions of our work:
>
> - Benchmark Design: AVI-Bench is **the first benchmark grounded in human cognition** to quantify the audio-visual capabilities of Omni-MLLMs. Our four-stage evaluation framework: Sensation, Perception, Understanding, and Reasoning, provides a unified and interpretable structure for benchmarking.
>
> - Four-Level Taxonomy and Metrics: The taxonomy evaluates intelligence across four dimensions: Task-, Modality-, Stage-, and Domain-Adaptive. It is **the first taxonomy to categorize human-like audio-visual intelligence in Omni-MLLMs**.
>
> - Insightful Findings:
> Across evaluations of ~30 models, we report seven novel and critical observations (e.g., modality imbalance, stage-specific bottlenecks) that **highlight current limitations and suggest promising directions for future research**.
>
> ---
> ### 2. Response to "... multimodality benchmarks cannot replace single-modality benchmarks"
> We fully acknowledge that single-modality benchmarks (e.g., MMMU [1] for image understanding, MMAU [2] for audio understanding) are essential and remain crucial for evaluating models within specific domains.
>
> However, the tasks and use cases addressed by Omni-MLLMs differ significantly from those focused on single-modality models. Omni-MLLMs are better suited to our world, which naturally includes both visual and auditory information. As multimodal models have garnered increasing attention in recent years, there is a growing need for a benchmark that assesses not only single-modal tasks (e.g., vision, audio) but also the interactions between modalities (e.g., audio-visual), which is precisely the insight that AVI-Bench aims to provide to the audio-visual research community. The goal of AVI-Bench is not to replace benchmarks like MMMU, but rather to complement them by offering a broader perspective on how well models integrate and reason across audio-visual modalities.
>
> Additionally, AVI-Bench fills a gap by focusing on cognitive principles, which allow us to identify not just performance, but the types of cognitive challenges and on what extent of human-like intelligence they have achieved that these models face when processing complex multimodal data. This ability to assess the underlying processes is a key feature that sets AVI-Bench apart from more traditional, modality-specific benchmarks.
>
> ---
> We hope these clarifications highlight the unique contributions of our work and its importance in advancing the field of Omni-MLLMs evaluation. We are grateful for your thoughtful comments and we will revise our Section `1. Introduction` and `Section 6. Conclusion` for a better contribution clarification.
>
> > [1] MMMU. arxiv.org/abs/2311.16502
> [2] MMAU. arxiv.org/abs/2410.19168

---

### Official Review · Reviewer_LNpg · 2025-11-01

**Soundness:** 3
**Presentation:** 3
**Contribution:** 3
**Rating:** 4
**Confidence:** 4

**Summary:**

This paper presents a novel benchmark for evaluating Omni-MLLMs, encompassing 14 diverse tasks including classification, grounding, matching, retrieval, QA, and captioning, among others. The benchmark offers a more comprehensive evaluation framework, structured around Perception, Understanding, Reasoning, and Primitive Sensation. Notably, the paper reports several interesting experimental findings, such as the observed discrepancy between audio and visual intelligence. In addition, it introduces innovative metrics like the modality-adaptive score, which is designed to assess such modality-specific discrepancies.

**Strengths:**

1. The experimental findings are interesting, particularly observation 3, which could inform future research aimed at improving audio support.
2. The four-level metrics introduced in Section 5 are interesting, providing a novel framework for evaluating models’ abilities in terms of balanced modality and stage performance.
3. The experiments are comprehensive and solid.

**Weaknesses:**

Lack of novelty: The task types in AVI-Bench are largely already covered by existing audio-visual benchmarks. Moreover, AVI-Bench omits several important audio-visual tasks, such as speech-speaker matching and conversation-action-camera interleaved captioning, which are critical for real-world applications of audio-visual models.

**Questions:**

None.

---

> ### Author Response · Authors · 2025-11-25
> **Response to Reviewer LNpg**
>
> We sincerely appreciate your positive evaluation of the **benchmark novelty, task diversity, contributions, and metric innovation**, as well as your recognition of our **solid and comprehensive experiments with interesting results**. Below, we address your concerns regarding novelty and task coverage. Should these responses adequately clarify your questions, we would greatly appreciate it if you could reconsider your evaluation.
>
> ---
> ## R1. On the concern that most task types already exist
>
> ### 1. Why most of the task types already exist
> - While AVI-Bench does incorporate existing task types, these **serve as foundational components rather than the core of our contribution**. The **selected tasks are classic, canonical, and thoroughly studied in the audio-visual community**, which is also `the foundations of many other existing audio–visual tasks`.
> - AVI-Bench does not aim to introduce new tasks or pursue exhaustive coverage of all tasks. Instead, we focus on exploring a unified and cognitively grounded framework to quantify Omni-MLLMs' human-like audio-visual intelligence.
>
> ### 2. Our efforts on dataset construction
> - Although some tasks appear in prior datasets, over 62% of AVI-Bench is newly annotated.
> - We format QA tasks with additioanl options, e.g., "same" or "not sure" to reduce the guessing and randomness.
> - We introduce a `response masking` mechanism (cf. Sec. A.2.2) to **improve evaluation reliability**.
> - For the Primitive Sensation level, we introduce the `audio–visual sensation QA (AVSQA)` task for the first time and construct its corresponding dataset.
>
> ### 3. Recall our contributions
> - Cognitively Inspired Evaluation Framework:
> Four-stage evaluation grounded in human cognition: Sensation, Perception, Understanding, and Reasoning, offering a unified and interpretable benchmark design.
> - Four-Level Intelligence Taxonomy:
> A taxonomy that evaluates Task-, Modality-, Stage-, and Domain-Adaptive intelligence, enabling principled quantification of Omni-MLLM capabilities.
> - Insightful Findings:
> Across evaluations of ~30 models, we report seven key observations (e.g., modality imbalance, stage-specific bottlenecks) that reveal current limitations and point toward future research directions.
>
> These contributions and novelty are also acknowledged by you (and we sincerely thank again). We will revise the descriptions in Section `1. Introduction` and `Section 6. Conclusion` to better emphasize our contributions.
>
> However, we agree with you that exploring new tasks is important for advancing audio–visual research. While not the focus of this work, we hope AVI-Bench inspires future designs of more challenging tasks and advances Omni-MLLMs.
>
> > [1] AV-Odyssey. arxiv.org/abs/2412.02611.
> [2] Q-bench. arxiv.org/abs/2309.14181.
>
> ---
> ## R2. On the concern of omitted tasks
> We greatly appreciate your forward-looking perspective. In the early stages of this project, we extensively discussed whether speech-related tasks should be included. Ultimately, we chose **NOT** to incorporate speech in the current version of AVI-Bench based on the following considerations:
>
> - While speech is conveyed through the audio modality, it primarily represents **linguistic information**. Evaluating speech-related tasks (e.g., ASR, speaker–speech matching, spoken conversation) largely tests a model’s semantic language understanding. We view language as a modality that can manifest through multiple channels, including speech (e.g., ASR), text in images (e.g., OCR), lip movement (e.g., VSR).
> - Although these tasks operate in different sensory modalities, **they are projections of the language, and primarily test a model’s ability to process linguistic content**.
>
> However, as we mentioned in the Broader Impacts (cf. Sec. A.9), we fully agree with that on the significance of tasks such as speaker-speech matching, which require integration of speech processing, visual identity recognition, and fine-grained cross-modal alignment. These are highly challenging and valuable directions for future research. Hence, we plan to develop a benchmark centered on language as a cross-modal bridge. This future benchmark will explore how linguistic content is expressed across various modalities, including speech, music, lip motion, hand sign, and text in images. And we further investigate modality-specific cues, such as:
> - Prosody and emotion embedded in speech;
> - Micro-expressions and individualized articulation patterns in lip movements;
> - Non-verbal emotional signals conveyed through music.
>
> We believe such extensions will significantly broaden the scope of AVI-Bench and lay the groundwork for unified modeling of language–perception–expression in future multimodal AGI systems.
>
> We sincerely thank you again for highlighting this meaningful direction. We will include a discussion on alternative audio modalities and their future integration in *Limitation* section in the revised version.

---

### Meta-Review · Area_Chair_m2E3 · 2026-01-06

**Summary:**

This paper proposed a new AV benchmark and also give insights are the shortages of current omni models. Concerns from reviewers are:

- Novelty and contribution positioning: many tasks reused from existing benchmarks

- Dataset scale vs. strength of claims: whether the scale supports claims related to human-like cognition/AVI

- Soundness of evaluation design and metrics: cross-task averaging, normalization, Eq. 1 / Eq. 2, definition of A/V, and stage aggregation/order

- Coverage and contextualization: missing important AV tasks, insufficient comparison with related benchmarks, and model coverage

**Reviewer Concerns:**

- Novelty and contribution positioning

Partially addressed, the author clarified the novelty. I agree with the reviewers that the novelty of this paper is quite limited.

- Dataset scale vs. strength of claims

Addressed. I agree with the authors that the review 3GeF is unreasonable on this point. Eval set is not designed to be super large.

- Soundness of evaluation design and metrics

Addressed: The rebuttal explains metric normalization, balanced task counts, precise definitions of A and V, handling of the A+V=0 case, and the rationale behind stage-adaptive aggregation.

- Coverage and contextualization

Partially addressed: Additional models are evaluated, and future extensions are outlined, but the benchmark scope itself remains intentionally constrained.

**Reviewer Scores:**

3GeF may change from 2 to 6 as multiple misunderstanding is classified

Other reviewers may keep the score as the concern regarding the novelty is valid, and hard to be changed during the rebuttal.

---

### Decision · Program_Chairs · 2026-01-26

Reject